# Prognostic Implications of MRI Melanin Quantification and Cytogenetic Abnormalities in Liver Metastases of Uveal Melanoma

**DOI:** 10.3390/cancers13112728

**Published:** 2021-05-31

**Authors:** Toulsie Ramtohul, Khadija Ait Rais, Sophie Gardrat, Raymond Barnhill, Sergio Román-Román, Nathalie Cassoux, Manuel Rodrigues, Pascale Mariani, Leanne De Koning, Gaëlle Pierron, Vincent Servois

**Affiliations:** 1Department of Radiology, Institut Curie, 75005 Paris, France; toulsie.ramtohul@curie.fr; 2Department of Genetics, Institut Curie, 75005 Paris, France; khadija.aitrais@curie.fr (K.A.R.); gaelle.pierron@curie.fr (G.P.); 3Department of Biopathology, Institut Curie, 75005 Paris, France; sophie.gardrat@curie.fr; 4Department of Translational Research, Institut Curie, 75005 Paris, France; raymond.barnhill@curie.fr (R.B.); sergio.roman-roman@curie.fr (S.R.-R.); leanne.de-koning@curie.fr (L.D.K.); 5Department of Surgical Oncology, Institut Curie, 75005 Paris, France; Nathalie.cassoux@curie.fr (N.C.); pascale.mariani@curie.fr (P.M.); 6Department of Medical Oncology, Institut Curie, 75005 Paris, France; manuel.rodrigues@curie.fr

**Keywords:** metastatic uveal melanoma, melanin quantification, prognosis, genetic classification

## Abstract

**Simple Summary:**

Melanin content in uveal melanoma is suspected to influence tumoral microenvironment and antitumoral response and is related to higher risk of metastasis and death in the primary disease. However, the prognostic impact of melanin content in liver metastases of uveal melanoma (LMUM) remains unexplored. The aim of our retrospective study was to evaluate the prognostic implications of melanin quantification assessed by MRI with clinical, pathological, and genetic features of LMUM. We found in a population of 63 patients eligible for margin-free resection of LMUM that MRI melanin quantification was an independent prognostic factor associated with overall survival. Liver metastases with high MRI melanin content and high genetic risk “M3/8g” were associated with lower overall survival compared with that of liver metastases with low MRI melanin content and/or low/intermediate genetic risk. The level of pigmentation in “M3/8g” LMUM identified two subsets that were correlated with distinct clinical outcomes.

**Abstract:**

To evaluate the prognostic implications of melanin quantification assessed by magnetic resonance imaging (MRI) with respect to the clinical, pathological, and genetic features of liver metastases of uveal melanoma (LMUM). This single-center retrospective cohort study included 63 patients eligible for margin-free resection of LMUM between 2007 and 2018. Comparative genomic hybridization of resected liver metastases on microarrays was performed for genetic risk classification. Metastases exhibiting monosomy 3 with any type of gain of chromosome 8 (M3/8g) were considered high-genetic-risk. MRI melanin quantification using the mean T1 signal (mT1s) in liver metastases was assessed quantitatively on preoperative imaging examination and compared to that of gross pathological evaluation. The association between MRI melanin quantification and overall survival (OS) was assessed by multivariate analysis using the Cox proportional hazards model. Gross pathological assessment of melanin content and MRI melanin quantification were strongly correlated (r = 0.8, *p* < 0.001). Independent prognostic factors associated with OS were disease-free interval ≤ 24 months (HR = 3.1; 95% CI, 1.6–6.0; *p* < 0.001), high-genetic-risk (HR = 2.2; 95% CI, 1.1–4.8; *p* = 0.04), mT1s > 1.1 (HR = 2.3; 95% CI, 1.2–4.7; *p* = 0.019), and complete hepatic resection (HR = 0.3; 95% CI, 0.2–0.7; *p* = 0.004). In patients with high-genetic-risk, mT1s showed a significant association with OS (HR = 3.7; 95% CI, 1.5–9.3; *p* = 0.006). The median OS was 17.5 months vs. 27 months for >1.1 and ≤1.1 mT1s tumors, respectively (*p* = 0.003). We showed that the level of pigmentation in M3/8g LMUM identified two subsets that were correlated with distinct clinical outcomes.

## 1. Introduction

Uveal melanoma (UM) is the most frequent primary malignancy of the eye in adults, and it exhibits molecular and pathologic features that differ from those of cutaneous melanoma [1]. Despite effective local treatment of the primary ocular tumor, metastatic spread occurs in 50% of patients, with the liver being the most common site [2]. The prognosis of liver metastatic disease remains very poor, with 1-year overall survival (OS) rates of approximately 50%. Various systemic treatments (chemotherapy, antiprogrammed cell death protein 1, target therapies, vaccine) were investigated in cases of liver metastases of uveal melanoma (LMUM) with disappointing results to date [3]. Recently, Tebentafusp, a bispecific fusion protein that can redirect T cells to target gp100+ cells, significantly improved OS compared to investigator’s choice in metastatic uveal melanoma, but only for patients with a specific HLA type [4]. For patients with oligometastatic liver involvement, invasive therapeutics, such as surgical resection of liver metastases, can be proposed [5]. For metastatic patients eligible for surgery, resection of LMUM is associated with the best OS rates published to date [6]. Liver involvement, LDH level and disease-free interval (DFI) between the primary ocular tumor and liver metastases were identified as independent prognostic factors of survival [7]. However, the prognostic impact of genetic alterations in LMUM is poorly explored, while genome profiling showing chromosomal aberrations (mainly monosomy 3 and 8q gain) or class 2 gene profiling (GEP) was the most significant prognostic factor of metastasis and survival at the local stage [8,9,10,11]. Additionally, melanin content in uveal melanoma is suspected to influence the tumoral microenvironment and the antitumoral response [12]. Higher pigmentation level in ocular uveal melanoma is related to a higher risk of metastasis and death [13,14]. Melanogenesis affects cellular metabolism, produces an oxidative environment, and has immunosuppressive properties that were linked to chemo- and radio-resistance in advanced melanoma [15]. Rothermel et al. identified hypopigmented metastasis as an immunogenic subset of LMUM that has robust antitumor reactivity and could benefit from immunotherapies that exploit endogenous antitumor T cell populations [16]. Also, tumor-infiltrating macrophage in particular proangiogenic M2-type which might be related to heavy pigmentation was associated with worse prognosis in ocular tumor [17,18]. However, the prognostic impact of melanin content in LMUM was not yet evaluated. Melanin content was pathologically correlated with hyperintensity in preoperative magnetic resonance imaging (MRI) T1-weighted images. The T1 shortening effect of melanin is likely based on the binding of paramagnetic metal ions [19].

The main goal of this study was to evaluate the prognostic implications of melanin levels, as assessed by MRI, and their association with the clinical, pathological, and genetic features of LMUM.

## 2. Results

### 2.1. Patient Characteristics

Between March 2007 and June 2018, among 165 metastatic patients in our surgical database, 63 patients eligible for margin-free resection of LMUM with available preoperative MRI, cytogenetic information and gross pathological melanin quantification were included. Eighteen of them were previously enrolled in two studies from the same institution [20,21]. The clinical, pathological, radiological, and genetic features and the management of the 63 analyzed patients are summarized in Table 1. The median age of the patients was 64 years (IQR: 48–70). The genomic classification of liver metastases included high-genetic-risk in 36 cases (57%) and low/intermediate-genetic-risk in 27 cases (43%), including 6 cases (10%) with D3/8nl, 3 cases (5%) with M3/8nl, and 18 cases (28%) with D3/8g. Treatment of primary ocular UM was proton beam or ^125^I–labeled plaque radiotherapy in 34 cases (54%) and enucleation in 29 cases (46%). The median disease-free interval (DFI) from UM to LMUM was 28 months (IQR: 13–57). The mean T1 signal (mT1s) was 0.95 (IQR: 0.76–1.22) for the whole cohort. Complete hepatic resection (R0) was performed for 51 (81%) patients. Postresection systemic treatment was administered to 51 (81%) patients.

### 2.2. Correlation of Gross Pathological Features and MRI Melanin Quantification

There were 13 (20.5%) hypopigmented, 23 (36.5%) mixed pigmented, and 27 (43%) hyperpigmented LMUM. The mT1s values were 0.67 (IQR: 0.64–0.68), 0.85 (IQR: 0.78–1.03), and 1.23 (IQR: 1.10–1.35) for hypopigmented, mixed pigmented, and hyperpigmented LMUM, respectively, and were significantly different (*p* < 0.001) (as illustrated in Figure 1). The intraclass correlation coefficient for inter-reader variability was 0.81 (95% CI, 0.72–0.89), corresponding to “almost perfect agreement”, according to the Landis and Koch guidelines [22]. Gross pathological assessment of melanin content and MRI melanin quantification were strongly correlated (r = 0.8, *p* < 0.001). Examples of images are provided in Figure 1. Also, gross pathological and immunohistochemical assessment of melanin content were significantly correlated (r = 0.7, *p* < 0.001) (as illustrated in Appendix A). There was no correlation between fundoscopic melanin evaluation of the primary tumor and the MRI melanin quantification in liver metastases (r = 0.1, *p* = 0.39). The best cutoff point of mT1s for distinguishing between hyperpigmented and hypopigmented/mixed pigmented LMUM, as assessed by the Youden index, was 1.1 with a sensitivity of 0.85, a specificity of 0.94, a predictive positive value of 0.90, and a predictive negative value of 0.92 (as illustrated in Appendix A). The mT1s was ≤1.1 in 36 patients (57%) and >1.1 in 27 patients (43%). Patients with mT1s LMUM >1.1 were more common in the low/intermediate pigmentation group (44% vs. 22%, *p* = 0.06).

### 2.3. Independent Prognostic Factors Associated with Survival

After a median follow-up of 70 months (IQR: 57–119), death occurred in 46 cases (73%), including 30 (83%) of the high-genetic-risk cases and 16 (59%) of the low/intermediate-genetic-risk cases. The median OS was 22.5 months for high-genetic-risk patients compared to 52 months for that of the low/intermediate-genetic-risk patients. According to multivariate Cox proportional hazards analysis, the independent prognostic factors associated with OS were DFI ≤ 24 months (HR = 3.1; 95% CI, 1.6–6.0; *p* < 0.001), high-genetic-risk (HR = 2.2; 95% CI, 1.1–4.8; *p* = 0.04), mT1s > 1.1 (HR = 2.3; 95% CI, 1.2–4.7; *p* = 0.019) and complete hepatic resection (HR = 0.3; 95% CI, 0.2–0.7; *p* = 0.004) (as illustrated in Table 2). There was no association with primary ocular characteristics, ocular treatment, or postresection systemic treatment. To evaluate the independent prognostic value of mT1s within DFI, genetic risk, and quality of liver resection, we performed univariate Cox proportional hazards analysis with OS as the end point (as illustrated in Appendix A). There was no association of mT1s with either DFI ≤ 24 or complete hepatic resection. However, there was a strong correlation of mT1s with genetic risk. Interestingly, within the group of patients with high-genetic-risk, there was a significant association between mT1s and OS (HR = 3.7; 95% CI, 1.5–9.3; *p* = 0.006) (as illustrated in Figure 2A). The median OS was 17.5 months vs. 27 months for tumors with mT1s >1.1 and ≤1.1, respectively (*p* = 0.003). Even within the subgroup of high-genetic-risk patients with complete hepatic resection (R0), low mT1s remained associated with longer median OS (18 vs. 40 months for >1.1 and ≤1.1 mT1s tumors, respectively (*p* = 0.01)) (as illustrated in Figure 2C). Similarly, within the subgroup of high-genetic-risk patients with DFI ≤ 24 months, low mT1s remained associated with longer median OS (14.5 vs. 27 months for >1.1 and ≤1.1 mT1s tumors, respectively (*p* = 0.04)) (as illustrated in Figure 2E). Among the patients with low/intermediate-genetic-risk, no significant association between mT1s and OS was found (HR = 1.4; 95% CI, 0.5–3.8; *p* = 0.54) (as illustrated in Figure 2B,D,F). The median OS was 45 months vs. 84 months for >1.1 and ≤1.1 mT1s tumors, respectively (*p* = 0.53).

To integrate genetic risk with MRI melanin classification, we defined three groups of patients (Group A: D3/8nl, D3/8g or M3/8nl abnormalities; Group B: M3/8g abnormalities and mT1s ≤ 1.1, and Group C: M3/8g abnormalities and mT1s > 1.1); these three groups of patients had distinct survival outcomes (*p* < 0.001) (as illustrated in Figure 3).

## 3. Discussion

To the best of our knowledge, this is the first study to investigate the prognostic value of melanin quantification for metastatic uveal melanoma. We found that in high-genetic-risk metastases, MRI quantification of melanin in liver metastases provided a significant prognostic information that was independent of genetic classification, DFI, and quality of liver resection.

Few studies explored prognostic factors for metastatic uveal melanoma. Valpione et al. found that percentage of liver involvement, serum LDH level, and WHO performance status were associated with worse prognosis [23]. In a collaborative European study, the performance index, the largest diameter of the largest metastasis and the alkaline phosphatase level at the time of diagnosis of metastases, were independent predictors of OS [7]. As in a retrospective study of 255 patients treated with liver surgery, we identified that DFI > 24 months and completeness of surgical resection (R0) were independent factors of survival [6]. However, LDH level, WHO performance status, DFI, quality of liver resection, and the size and number of liver metastases were not specific to metastatic uveal melanoma disease, and may reflect only the general health and the liver tumor burden of these patients. Additionally, none of the cited prognosis studies considered genomic abnormalities, which appeared to be a major feature at the primary ocular stage [24]. We recently published a nomogram based on 224 metastatic patients showing that a disease-free interval of less than 6 months, the presence of more than 10 LMUMs, largest LMUM more than 1200 mm^2^ in size, and lactate dehydrogenase (LDH) value greater than 1.5 were independently associated with worse outcomes and with acceptable predictive performance. Due to the lack of genomic data for a large number of these metastases, we were not able to demonstrate an impact of genetic alterations on survival in that study [21]. In a recent study of 41 UM liver metastases, where most patients had a high-genetic-risk, only the replacement histopathological growth pattern and the quality of liver resection were independent factors related to OS. The genomic high-risk variable M3/8g had no prognostic value at this stage of liver metastasis, possibly due to the small number of patients [20].

In the present study, we have complete genetic data for 63 liver samples, which show that monosomy 3 and chromosome 8 gain status are important prognostic factors for survival at the metastatic stage regardless of the patient’s baseline characteristics and his or her surgical management. Somatic alterations with inactivation of *BAP1* were identified in 81% of M3/8g UM and provided no additional prognostic value that was independent of cytogenetic risk classification. More than half of the patients in our cohort displayed M3/8g cytogenetic alterations in liver metastasis, and their median OS was half as that of the low/intermediate-genetic-risk group.

Also, we found that high-genetic-risk LMUM tended to have lower melanin pigmentation as assessed by MRI T1 signal, while low/intermediate-genetic-risk LMUM exhibited higher T1 signals. The expression of melanin pigment correlates negatively with the OS rate and the antitumoral response in patients with primary UM and cutaneous melanoma [12,25]. While both of these types of tumor arise from the common melanocyte lineage, two studies found that UM metastases displayed higher pigmentation scores than cutaneous melanoma metastases [16,26]. Melanogenesis is related to significant upregulation of hypoxia-dependent pathways, which contribute to producing a more aggressive phenotype of melanoma [27]. Recent integrative analyses revealed that hypoxia signaling is a major transcriptional signature of specific subsets of M3/8g uveal melanoma with poor prognosis [10,11]. Interestingly, we discovered that M3/8g LMUM with poor prognosis can be subdivided into two subsets based on melanin MRI quantification that correlated with distinct clinical outcomes (18 months vs. 30 months for mT1s > 1.1 and ≤ 1.1 LMUM, respectively). When surgical resection was optimal (R0 subgroup), the OS reached 40 months for M3/8g tumors with poor prognosis and mT1s ≤ 1.1. These data show that important disparities in survival rates exist among subgroups of metastatic UM patients and that survival rates can be significantly more favorable than the median OS of 13 months described in a recent meta-analysis of 2494 patients across all treatment modalities [28]. For patients with low/intermediate-genetic-risk, there was a trend toward better OS for mT1s ≤ 1.1 metastases; this trend did not reach statistical significance, however, potentially due to the lower rate of events in this group. The median OS reached 84 months for patients with tumors with mT1s ≤ 1.1, indicating that an invasive strategy might be the best therapeutic option in this group.

Two studies of the evolution of metastatic uveal melanoma found limited genetic heterogeneity with no new actionable driver mutations in the metastases and very few new mutations associated with tumor progression [29,30]. Therefore, genetic risk based on primary tumor samples after enucleation or fine-needle aspiration at the time of plaque positioning could be used to estimate the cytogenetic alterations in liver metastases. These findings could have several important implications in clinical practice. Indeed, noninvasive MRI quantification of melanin in liver metastases combined with analysis of cytogenetic alterations in primary ocular UM may serve for decision-making and risk stratification for therapeutic options. For example, mT1s ≤ 1.1 high-genetic-risk patients eligible for margin-free resection may benefit from liver surgery with promising median OS, whereas mT1s > 1.1 high-genetic-risk patients who exhibit the poorest outcomes might benefit from systemic treatments in phase 1 clinical trials.

Our study has some limitations; one of these is its retrospective monocentric design, which could give rise to selection bias. First, only patients amenable to surgery were included. Second, the large proportion of patients with small metastases and limited liver tumor burden may reflect the aggressive screening strategy with MRI, especially for high-genetic-risk patients, since early MRI diagnosis of liver metastasis of uveal melanoma might select appropriate candidates for complete surgical resection [21].

## 4. Material and Methods

### 4.1. Study Population

This retrospective study was approved by the Uveal Melanoma Institutional Review Board of Institut Curie (MU-10-2018). Written informed consent for the use of tissue samples and data for research was signed by each patient. Between March 2007 and June 2018, we identified all patients eligible for margin-free resection (R0) of LMUM at the Institut Curie Hospital (Paris, France) with available preoperative MRI, cytogenetic analysis, and gross pathological melanin quantification. The following clinical data were recorded at the time of diagnosis of liver metastasis: patient age, sex, pretreatment ocular largest basal diameter (LBD) and tumor thickness of UM, ciliary body involvement, fundoscopic melanin evaluation of UM, ocular treatment modalities, DFI between UM and LMUM, number of metastases and largest metastasis size, presence or absence of miliary disease, MRI T1 signal of liver metastases, cytogenetic alterations and somatic mutations in liver metastases, postresection systemic treatment, and date of death or of last followup. Miliary disease was defined by the presence on liver MRI of at least three liver metastases measuring less than 5 mm. After liver resection, the systemic treatments administered for relapse or incomplete surgical resection (R2) were immunotherapy (pembrolizumab, nivolumab), chemotherapy (fotemustine, dacarbazine, cyclophosphamide, sorafenib), or targeted therapy as part of phase 1 clinical trials. The study complied with the principles set forth in the Declaration of Helsinki. This report was written in accordance with STARD guidelines [31].

### 4.2. Genetic Analysis

Written informed consent to perform somatic genetic analyses was obtained from all patients. Comparative genomic hybridization on microarrays (array-CGH) was performed on resected liver metastases as previously described [24]. A detailed technical protocol is provided in the Appendix A. The specimens were reviewed by two expert geneticists (KA and GP) with 10 years of expertise (GP) in the field of melanoma. According to previous prognostic correlation studies, genomic profiles were classified based on chromosome 3 and 8 status. Metastases exhibiting monosomy 3 with any type of gain of chromosome 8 (M3/8g) were considered high-genetic-risk. Metastases exhibiting disomy 3 with normal dosage or gain of chromosome 8 (D3/8nl or D3/8g), monosomy 3, or isodisomy with normal dosage of chromosome 8 (M3/8nl) were considered low/intermediate-genetic-risk. Patients were classified according to the cytogenetic alterations found on examination of resected liver samples.

### 4.3. Gross Pathological and Histopathological Assessment of Pigmentation in Liver Metastases

Postoperative diagnosis of LMUM was performed according to a combination of well-defined pathological and immunohistochemical criteria by expert pathologists (SG and RB) with 37 years of expertise (RB) in the field of melanoma. Melanin quantification was assessed visually by consensus using digitized images of the excised tissues and was graded as follows: 0+ (hypopigmented), 1+ (mixed pigmented), and 2+ (strongly pigmented), as described by Rothermel et al. [16]. Details of immunohistochemical examination of liver metastases for melanin content utilizing the melanin index are provided in the Appendix A.

### 4.4. MRI Melanin Quantification in Liver Metastases

Preoperative MR imaging examinations were performed on a 1.5-T clinical MR scanner (Symphony, Siemens, Erlangen, Germany). The detailed MRI protocols are described in Appendix A. Melanin MRI quantification was assessed quantitatively based on analysis of unenhanced 3D dual gradient echo T1-weighted in phase sequence by an abdominal radiologist (VS) with 25 years of experience in abdominal MRI. The radiologist was aware of the liver segment of the resected metastases but was blinded to the baseline characteristics and oncologic outcomes. Regions of interest (ROIs) were manually drawn in tumors and in the adjacent liver parenchyma, and the sizes of these ROIs were arbitrarily chosen to be between 100 and 200 mm^2^, avoiding liver lesions or vessels. The mean T1 signal (mT1s) was defined as the ratio of the signal from the liver metastasis to that of the normal surrounding liver parenchyma on unenhanced 3D dual gradient echo T1-weighted images in phase. To provide an assessment of inter-reader variability, a second abdominal radiologist (TR) with 6 years of experience independently performed a second set of MRI melanin quantifications following the same protocol.

### 4.5. Study Endpoints

The main outcome was OS, defined as the interval from liver resection to death due to any cause or last patient contact. Patients who were alive on the cutoff date (1 July 2019) were censored at the last assessment date.

### 4.6. Statistical Analysis

Continuous variables were analyzed using the Mann–Whitney and Student tests according to distribution normality. Categorical variables were analyzed using the χ2 test or Fisher’s exact test. The Kaplan–Meier product-limit method was used to plot time-to-event outcomes, and comparisons between curves were performed with the exact log-rank test. The best cutoff point of mT1s to distinguish between hyperpigmented (2+) vs. hypopigmented (0+)/mixed (1+) pigmented LMUM was assessed by Youden index. The association between mT1s and OS was assessed by multivariate analysis using the Cox proportional hazards model, entering all other variables at the *p* < 0.2 level in univariate analysis and then applying backward selection to retain significant factors at *p* < 0.05 in the model (DFI, mT1s, genetic classification and quality of liver resection). Schoenfeld residuals were used to check the proportional hazard assumption. The inter-reader variability in mT1s between observers was assessed using the intraclass correlation coefficient. All analyses were performed using SAS software (Version 9.4, SAS Institute, Cary, NC, USA). All statistical tests were two-sided, and *p*-values less than 0.05 were considered statistically significant.

## 5. Conclusions

Our results suggest that the prognosis of oligometastatic uveal melanoma depends on hepatic features such as DFI, quality of liver resection, genetic risk classification, and melanin quantification in liver metastases. We found that the level of pigmentation as assessed by MRI identified two subsets of M3/8g LMUM that correlated with distinct clinical outcomes. These findings could have important implications in risk stratification for therapeutic options.

## Figures and Tables

**Figure 1 cancers-13-02728-f001:**
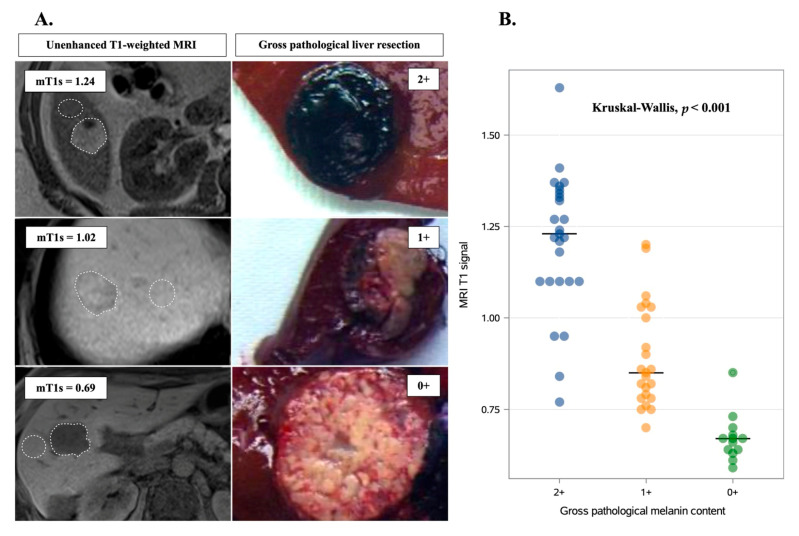
Association between gross pathological and MRI melanin quantification. (**A**) Illustrative examples showing the correlation between preoperative, unenhanced, T1-weighted MRI and postoperative gross pathological scoring of pigmentation in uveal melanoma liver metastases. White dashed lines indicate regions of interest that were manually drawn in tumors and in adjacent liver parenchyma. Melanin pathological quantification was assessed visually and graded as follows: 0+ (hypopigmented), 1+ (mixed pigmented), and 2+ (strongly pigmented). (**B**) Correlation of gross pathological features and MRI melanin quantification. Horizontal lines indicate median values.

**Figure 2 cancers-13-02728-f002:**
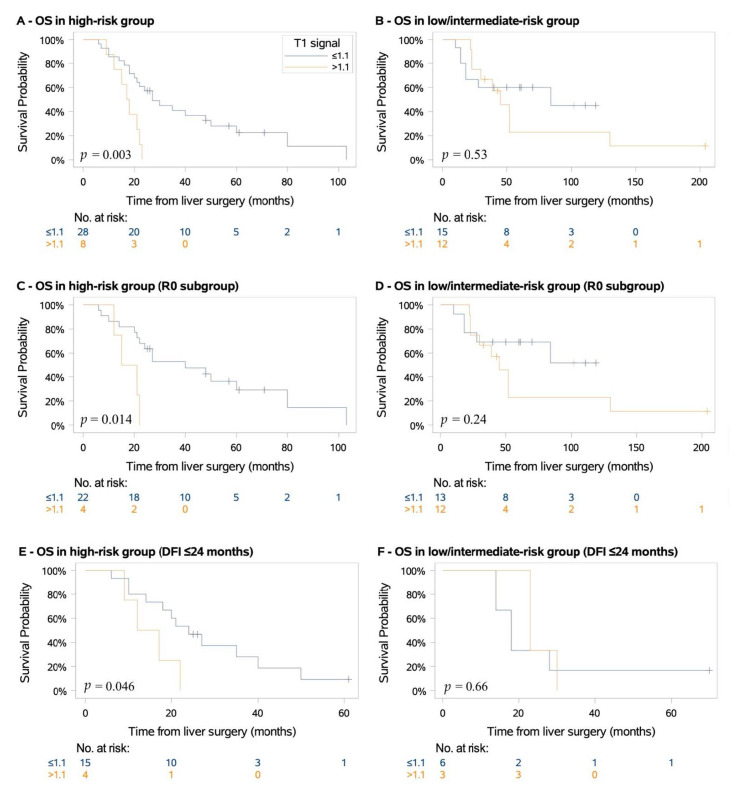
Overall survival (OS) of patients with metastatic hepatic uveal melanoma. *p* values were calculated using log-rank test. R0, complete hepatic resection; DFI, disease-free interval. (**A**) overall survival in high-genetic-risk group, (**B**) overall survival in low/intermediate-genetic-risk group, (**C**) overall survival in high-genetic-risk group with complete hepatic resection (R0), (**D**) overall survival in low/intermediate-genetic-risk group with complete hepatic resection (R0), (**E**) overall survival in high-genetic-risk group with disease-free-interval ≤ 24 months, (**F**) overall survival in low/intermediate-genetic-risk group with disease-free-interval ≤ 24 months.

**Figure 3 cancers-13-02728-f003:**
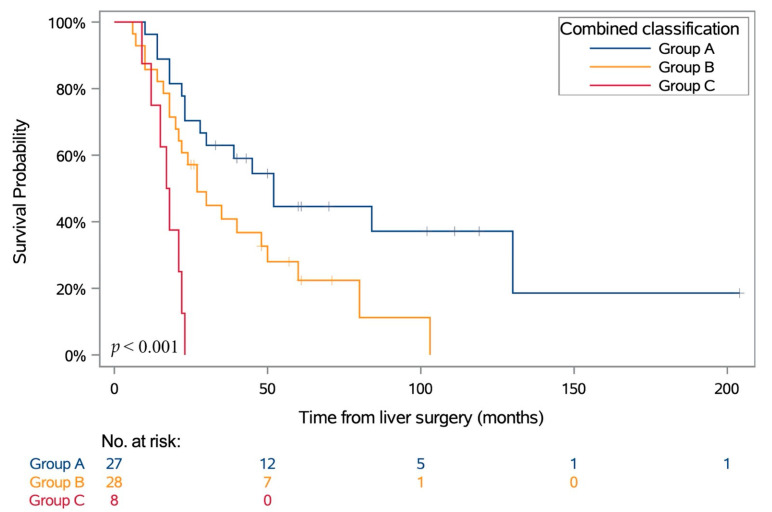
OS according to combined classification based on genetic risk and MRI melanin quantification. *p* values were calculated using log-rank test. Group A refers to D3/8nl, D3/8g or M3/8nl; Group B refers to M3/8g abnormalities and mT1s ≤ 1.1; Group C refers to M3/8g abnormalities and mT1s > 1.1.

**Table 1 cancers-13-02728-t001:** Clinical, pathological, genetic, and radiological features of metastatic hepatic uveal melanoma cohort at liver resection operation.

Baseline Characteristics			Genetic Classification	
Covariate	Label	All Cohort	High-Risk (*n* = 36)	Low/Intermediate-Risk (*n* = 27)
Age (years)	≤60	29 (46%)	14 (39%)	15 (56%)
	>60	34 (54%)	22 (61%)	12 (44%)
Sex	Female	30 (48%)	17 (47%)	13 (48%)
	Male	33 (52%)	19 (53%)	14 (52%)
Largest basal diameter (mm) of primary tumor	≤15	43 (68%)	21 (58%)	22 (81%)
	>15	20 (32%)	15 (42%)	5 (19%)
Ciliary involvement of primary tumor	No	53 (84%)	27 (75%)	26 (96%)
	Yes	10 (16%)	9 (25%)	1 (4%)
Retinal detachment of primary tumor	No	41 (65%)	23 (64%)	18 (67%)
	Yes	22 (35%)	13 (36%)	9 (33%)
Fundoscopic melanin evaluation of the primary tumor	NA	4 (7%)	0 (0%)	4 (15%)
	Absent	2 (3%)	1 (3%)	1 (4%)
	Moderate	14 (22%)	7 (19%)	7 (26%)
	Heavy	43 (68%)	28 (78%)	15 (55%)
Ocular tumor treatment	Enucleation	29 (46%)	20 (57%)	9 (32%)
	Proton beam/^125^I-plaque	34 (54%)	15 (43%)	19 (68%)
Disease-free interval between primary tumor and liver metastases (months)	≤24	28 (44%)	19 (53%)	9 (33%)
	>24	35 (56%)	17 (47%)	18 (67%)
Size of liver metastases (mm)	≤20	45 (71%)	28 (78%)	17 (63%)
	>20	18 (29%)	8 (22%)	10 (37%)
Number of liver metastases	≤4	49 (78%)	29 (81%)	20 (74%)
	>4	14 (22%)	7 (19%)	7 (26%)
Miliary disease *	No	27 (43%)	11 (31%)	16 (59%)
	Yes	36 (57%)	25 (69%)	11 (41%)
T1 signal	≤1.1	43 (68%)	28 (78%)	15 (56%)
	>1.1	20 (32%)	8 (22%)	12 (44%)
Histology type of liver metastases	Spindle	19 (30%)	9 (25%)	10 (37%)
	Epithelioid-mixed	44 (70%)	27 (75%)	17 (63%)
Loss of *BAP1* of liver metastases	NA	6 (10%)	3 (8%)	3 (11%)
	No	23 (37%)	5 (14%)	18 (67%)
	Yes	34 (54%)	28 (78%)	6 (22%)
Quality of liver resection	R0	51 (81%)	26 (72%)	25 (93%)
	R2	12 (19%)	10 (28%)	2 (7%)
Number of therapeutic lines after liver resection	≤2	37 (59%)	20 (56%)	17 (63%)
	>2	26 (41%)	16 (44%)	10 (37%)
Systemic treatment	No	12 (19%)	3 (8%)	9 (33%)
	Yes	51 (81%)	33 (92%)	18 (67%)
Chemotherapy	No	18 (29%)	8 (22%)	10 (37%)
	Yes	45 (71%)	28 (78%)	17 (63%)
Immunotherapy	No	27 (43%)	12 (33%)	15 (56%)
	Yes	36 (57%)	24 (67%)	12 (44%)

Data are expressed as *n* (%). High-genetic-risk refers to M3/8g abnormalities, and low/intermediate-risk refers to D3/8nl, D3/8g, or M3/8nl. * Miliary disease was defined radiologically by presence of at least three liver metastases of less than 5 mm. NA: not available.

**Table 2 cancers-13-02728-t002:** Multivariate Cox proportional hazards analysis of overall survival.

Covariate	Reference	Hazard Ratio (95% CI)	*p*-Value
Disease-free interval (months)	≤24	3.1 (1.6–6.0)	<0.001
>24			
T1 signal	≤1.1	0.4 (0.2–0.9)	0.019
>1.1			
Genetic classification	High-risk	2.2 (1.0–4.8)	0.040
Low/intermediate-risk			
Quality of liver resection	R0	0.3 (0.2–0.7)	0.004
R2			

For multivariate analysis, a backward selection approach was performed starting from all variables significant in univariate analysis at the *p* < 0.2 level. R0, complete hepatic resection; R2, incomplete hepatic resection.

## Data Availability

The data presented in this study are available on request from the corresponding author.

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
