# Peer review of "Prognostic Implications of MRI Melanin Quantification and Cytogenetic Abnormalities in Liver Metastases of Uveal Melanoma"

_cancers, 2021, doi:10.3390/cancers13112728_

Round 1

Reviewer 1 Report

Nice and valuable article. 

My only 3 main points are:

  • the manuscript is about the signal intensity of lesion on MRI. I therefore think it is important to show some of these images in the main manuscript, eg a hyperintense and hypointense lesion, and the ROIs which were drawn.
  • How is the cutoff of 1.1 determined? I would personally expect a cutoff of 1.0 since you compare the lesion to healthy tissue.
  • You provide quite some details about the primary UM. It would be valuable to also provide information about the pigmentation and assess whether this matched the signal intensity of the metastasis.

Minor points:

  • line 114: T1-weighted imaging  should be T1-weighted images.
  • line 117: same rules should be same protocol.

Reviewer 2 Report

Authors evaluate the prognostic implications of melanin quantification assessed by magnetic resonance imaging (MRI) with respect to the clinical, pathological and genetic features of LMUM. They concluded that the level of pigmentation in LMUM identified two subsets that were correlated with distinct clinical outcomes The study is well written but it can be improved including some minor revisions:

1) Include H&E stained sections of LMUM with different level of pigmentation

2) Include depigmented stained sections of LMUM

3) Indicate number of epithelioid and spindle cells LMUM 

Reviewer 3 Report

This is a short, concise manuscript that reports preoperative magnetic resonance imaging (MRI) quantification of melanin in uveal melanoma liver metastases (mUM) from 63 patients.  Melanin quantitative data was correlated with genetic risk based on chromosome 3 and 8 status of both mUMs and primary tumors (pUMs), gross morphological assessment, quality of liver resection, and survival.  High genetic risk mUMs were generally found to have lower melanin content than lower genetic risk mUMs. Using a multivariate analysis that included melanin quantification, the authors identified two high genetic risk mUMs subgroups (n=36 patient), with one group (n=28 patients)  exhibiting significantly longer survival times (~4x) following liver surgery than the other group (n=8 patients).   The authors propose that non-invasive MRI melanin measurements could assist clinicians in deciding therapeutic approaches for poor prognosis patients (eg, surgery vs systemic clinical trial treatments).  These authors have the required expertise for this research(2018 J Pathol Clin Res 4, 227; 2019 Cancers 11, 863) and the results in this report appear sound.  Nevertheless the patient sample size in this study is small and more work will be required to validate the prognostic benefits of melanin quantification.

Style wise this report does not adhere to the Instructions to the Authors:  a Simple Abstract is missing and Materials and Methods is presented before the Results rather than after the Discussion.  Abbreviations in the abstract should be limited to the minimum and defined when used.

Reviewer 4 Report

The manuscript entitled “Prognostic implications of MRI melanin quantification and cy-2 togenetic abnormalities in liver metastases of uveal melanoma” addresses the interesting issue of differential melanin presence in liver metastases of uveal melanoma and its prognostic potential.

The study is well planned and conducted, the study population is impressive given the nature of the disease and the associated numbers. The manuscript is well written, data are well reported and correctly interpreted.

Major remarks:

High genetic risk is defined by disomy of chromosome 3 and amplification of chr8q. Most high risk cases are likely to show chr8q and almost all are expected to show chr3 monosomy. Chr8q amplification has been described to be associated with a major infiltrate in the primary tumor. It would be therefore important to verify by IHC whether the same holds true for LMUM and whether the infiltrate is associated with different mT1s status. In parallel, existing gene expression data should be used to verify whether chr8q fain is associated with melanin expression and whether the latter is associated with extent or type of infiltrate.

Ethics: The authorization from the institutional board must be specified, information on informed consent must be given

Minor remarks:

Abstract:

Spell out/define LMUM, DFI, M3/8g

Briefly define “high genetic risk”

“liver uveal melanoma (UM) metastases” should read “liver metastases of uveal melanoma (UM)”

Introduction

Among the treatment opportunities cited recent success with Tebentafusp should be discussed.

The discussion of hypopigmented immunogenic metastases should be extended by inclusion of the type of infiltrate observed (pro- / anti-tumoral).

Genetic risk of metastasis should be introduced and reference to recent reviews on the subject should be given.

Methods

“Tumors 90 exhibiting monosomy 3 with any type of gain of chromosome 8 (M3/8g)” clarify whether this refers to primary tumors or not.

Within the low genetic risk group the proportion of D3/8nl and D3/8g versus M3/8nl should be indicated.

Results

It is not clear whether the tumor features reported in the text and in table 1 refer to primary tumors or to liver metastases.

Indicate the numbers of D3/8nl, D3/8g or M3/8nl abnormalities.

Discussion

“twice as low” should read “half”

Author Response

Response to Reviewer 4 Comments

Point 1: High genetic risk is defined by disomy of chromosome 3 and amplification of chr8q. Most high risk cases are likely to show chr8q and almost all are expected to show chr3 monosomy. Chr8q amplification has been described to be associated with a major infiltrate in the primary tumor. It would be therefore important to verify by IHC whether the same holds true for LMUM and whether the infiltrate is associated with different mT1s status. In parallel, existing gene expression data should be used to verify whether chr8q fain is associated with melanin expression and whether the latter is associated with extent or type of infiltrate.

Response 1: We thank the reviewer for raising these important points. However, the comprehensive analysis and correlation of immune cell infiltrates with genetic parameters in uveal melanoma is the subject of another manuscript in preparation and not the subject of this manuscript.  

Point 2: The authorization from the institutional board must be specified, information on informed consent must be given.

Response 2: We added the details of the authorization from the institutional board:” This retrospective study was approved by the institutional ethics review board (MU-10-2018). Written informed consent for the use of tissue samples and data for research was signed by each patient.” (line 372-374). Also, please find attached the INFORMATION ON COLLECTION AND USE OF YOUR BIOLOGICAL SAMPLES

Point 3: Spell out/define LMUM, DFI, M3/8g. Briefly define “high genetic risk” ; “liver uveal melanoma (UM) metastases” should read “liver metastases of uveal melanoma (UM)”.

Response 3: We added a definition of the abbreviations in the abstract

Point 4: Among the treatment opportunities cited recent success with Tebentafusp should be discussed.

Response 4: We added in the introduction the recent success with Tebentafusp : “Recently, Tebentafusp, a bispecific fusion protein that can redirect T cells to target gp100+ cells, significantly improved OS compared to investigator’s choice in metastatic uveal melanoma but only for patients with a specific HLA type.” (line 99-102)

Point 5: The discussion of hypopigmented immunogenic metastases should be extended by inclusion of the type of infiltrate observed (pro- / anti-tumoral).

Response 5: We added a discussion for the type of immune infiltrate: “Also, tumor-infiltrating macrophage in particular pro-angiogenic M2-type which might be related to heavy pigmentation has been associated with worse prognosis in ocular tumor.” (line 118-120)

Point 6: Genetic risk of metastasis should be introduced and reference to recent reviews on the subject should be given.

Response 6: Genetic risk of metastasis and recent reviews of molecular subset of UM were added:while genome profiling showing chromosomal aberrations (mainly monosomy 3 and 8q gain) or class 2 gene profiling (GEP) was the most significant prognostic factor of metastasis and survival” (line 109-110)”

Point 7: “Tumors 90 exhibiting monosomy 3 with any type of gain of chromosome 8 (M3/8g)” clarify whether this refers to primary tumors or not.

Response 7: The genetic analysis and classification refer to analyses performed on resected liver metastases. We clarified the genetic analysis section as follows: “Comparative genomic hybridization on microarrays (array-CGH) was performed on resected liver metastases as previously described” (line 392-393)

Point 8: Within the low genetic risk group the proportion of D3/8nl and D3/8g versus M3/8nl should be indicated. Indicate the numbers of D3/8nl, D3/8g or M3/8nl abnormalities.

Response 8: The respective number of D3/8nl, D3/8g or M3/8nl abnormalities was added in the text: “low/intermediate-genetic-risk in 27 cases (43%) including 6 cases (10%) with D3/8nl, 3 cases (5%) with M3/8nl and 18 cases (28%) with D3/8g.” (line 138-139)

Point 9: It is not clear whether the tumor features reported in the text and in table 1 refer to primary tumors or to liver metastases.

Response 9: We clarified the text and the Table 1 to better distinguish between primary ocular UM features and liver metastases of UM features.

Point 10: “twice as low” should read “half”

Response 10: We thank the reviewer for the language style change proposition: “their median OS was half as that of the low/intermediate-genetic-risk group.” (line 325-326)

Reviewer 5 Report

Thank you for this interesting article dealing with a large series of well-documented liver metastases from uveal melanoma. The main drawback of the paper is not to be focused enough, with many data and a discussion which is too difficult to follow. In the introduction the authors state that the main goal of the study is to evaluate the prognostic implications of melanin levels, as assessed by MRI. in the discussion they start with a sentence saying that the level of melanin assessed with MRI is strongly associated with pathological melanin assessment, which does not really answer the main question of the introduction. As a whole, the paper in the discussion could be better structured because it is somewhat difficult to read. There are many data, some of  which are not  put  at the forefront consistently throughout the manuscript. The organization of the discussion , if not the introduction, should be reviewed.

As far as the pathology is concerned, the words gross morphological assessment sounds weird - Moreover, the quantifiaction that has been performed is rather a semi quantification, as  melanin content was evaluated using scores, moreover without analysis of the intereader variability. A proper quantification of melanin content which means precise density measurements would and should have been performed using computerized image analysis. Why didn't the authors perform this analysis as they have the digitized images of the excised tissues? If the main question of this study is the correlation between assessment of melanin content with MRI and the gold standard represented by path, this should have be done. ImmunoHistochemical criteria would probably be a better word than immunophenotypical criteria

Could you please add the following refrences after the sentence line 162 page 5: Interobserver agreement issues in radiology.

Benchoufi M, Matzner-Lober E, Molinari N, Jannot AS, Soyer P. Diagn Interv Imaging. 2020 Oct;101(10):639-641. doi: 10.1016/j.diii.2020.09.001  Results section it is not quite clear what the results of the univariate analysis was ( p values for the criteria? could you please provide them, at least those of the variables that were included in the backward selection? Moreover, it is surprising that as much as 15 variables were kept for the mutlivariate analysis as the population is rather small (63 patients). Discussion Line  225 the sentence " we confirmed that the prognosis of metast. UM  was independent of primary ocular characteritics and management is not very clear. The quality of hepatic resection and the completeness of the resection seems to pertain to the management. maybe the authors mean managment of the ocular mangement, but this sentence should be clarifiesd and does not bring any information related to the main objectives of this paper so myabe it should be skipped as it only induces confusion. instead of lines 231 to 235, state what are the independent factors associated with outcome that are consistent through your study and theirs, and in case of discrepancies try and explain why there may be a discrepancy. Moreover, the sentence" however, these variables were not specific to metatstatic uveak melanom disease etc.." is not clear, please clarify. What variables? have they been tested in the present study, and if yes with what results? what do you mean by " at the local stage"? in the series you recently published, did they include some of the patients in this study? if yes, please state so in the material and methods section. Why didn't you try  include the same general variables in your univariate analysis? This d requires some clarification As correlating the melanin content assessed with MRI to the melanin content with path is an important issue of your paper, you should comment on your results with regard to those of the literature in the discussion.   Conclusion do not follow the main results presented at the beginning of the discussion   Figures could you please provide an example of an MR image and a corresponding macroscopical image to illustrate the correlation between both techniques with respect to the assessment of melanin content? References ok please just add the reference relative to the interobserver agreement cited above

Round 2

Reviewer 1 Report

  • The explanation of the 1.1 threshold of the MR-signal in the author response is good. This should however also be included in the manuscript, since it is an important part of the methodology.
  • regarding the point of the pigmentation of the primary UM (point 3), I do not fully agree with the response. Although a histological evaluation of the primary UM would indeed be the best, a funduscopic evaluation should be available for all patients, since this is needed for the diagnosis. I therefore still think this evaluation is feasible and should be included in the manuscript.

Reviewer 4 Report

The authors have adequately addressed the issues raised althpough I would have liked to see the molecular analysis in this manuscript. Yet I understand that the authors wish to prepare a separate manuscript on this. 

Author Response

We thank the reviewer for his comments

Reviewer 5 Report

Thank you for the revisions which are satisfactory

Author Response

We thank the reviewer for his comments

Round 3

Reviewer 1 Report

The authors have successfully improved the manuscript.